

# A low-cost method for monitoring snow characteristics at remote field sites

Rosamond J. Tutton[1], Robert G. Way[1]

[1]Department of Geography and Planning, Queen's University, Kingston, ON K7L 3N9, Canada

*Correspondence to*: Robert G. Way (robert.way@queensu.ca)

**Abstract.** The lack of spatially distributed snow depth measurements in natural environments is a challenge worldwide but particularly in northern regions such as coastal Labrador where changes to snow conditions directly impact indigenous livelihoods, local vegetation, permafrost distribution and wildlife habitat. This problem is exacerbated by the lack of cost-efficient and reliable snow observation methods available to researchers studying cryosphere-vegetation interactions in remote regions. In this study, we propose a new method termed snow characterization with light and temperature (SCLT) for estimating snow depth using vertically arranged multivariate (light and temperature) data loggers. To test this new approach, six snow stakes outfitted with SCLT loggers were installed in forested and tundra ecotypes in Arctic and Subarctic Labrador. The results from one-year of field measurement indicate that daily maximum light intensity (lux) at snow covered sensors is diminished by more than an order of magnitude compared to uncovered sensors. This contrast enables differentiation between snow coverage at different sensor heights and allows for robust determination of daily snow heights throughout the year. Further validation of SCLT is needed to resolve ambiguities with thresholds for snow detection and to elucidate the impacts of snow density on retrieved light and temperature profiles. However, the results presented in this study suggest that the proposed technique represents a significant improvement over prior methods for snow depth characterization at remote field sites in terms of practicality, simplicity, and versatility.

## 1 Introduction

Snow cover and snow depth are among the Global Climate Observing System's (GCOS) essential climate variables (Bojinski et al., 2014) and are critical components of global and regional energy balances (Olsen et al., 2011; Pulliainen et al., 2020). The global snow albedo effect influences all humans, but consequences of changing snow conditions for those living in cold climate and alpine regions are especially pronounced (Ford et al., 2019; Lemke et al., 2007). Accurate characterization of snow depth is important for hydroelectric operations, freshwater and land resource availability to communities and prediction of climate change impacts (Hovelsrud et al., 2011; Mortimer et al., 2020; Sturm et al., 2005; Thackeray et al., 2019; Wolf et al., 2013). Changes to snow depth and snow cover duration in Arctic and alpine tundra caused by enhanced shrub and tree growth can result in warmer ground temperatures, permafrost thaw and further vegetation expansion (Callaghan et al., 2011; Wilcox et al., 2019). Unlike its liquid counterpart, snow is hard to catch, melts differentially (Archer, 1998) and is



structurally, mechanically and thermally anisotropic (Leinss et al., 2016). Our ability to monitor *in situ* snow conditions has historically been limited to open areas near larger communities and airfields where large meteorological apparatus are established (Goodison, 2006). As such, standardized measurement of snow remains a challenge in remote regions where existing stations cannot represent the diversity of snow conditions across topography and vegetation (Brown et al., 2012, 2003; Derksen et al., 2014).


Satellite remote sensing platforms are unable to directly measure snow depth and thermal properties in most environments (Boelman et al., 2019; Kinar & Pomeroy, 2015; Sturm, 2015) and depend on a very limited network of surface validation sites located in open areas (Trujillo and Lehning, 2015). Further, acquisition, establishment and maintenance of stationary weather instrumentation used by government and industry services is costly outside of regional centres, and this

infrastructure is not designed to represent forest conditions (Goodison, 2006). This leads to data-sparse areas at high latitudes and in mountainous regions, and spatially biased representation of snow characteristics in research and modelling which reduce our ability to predict impacts of climate change on snow and ground conditions (Domine et al., 2019; Pulliainen et al., 2020).

To compensate for the lack of automated, spatio-temporal measurements, field researchers in ecological, hydrological and cryospheric domains have made use of low-cost methods such as vertically arranged temperature loggers (Gilbert et al.,

2017; de Pablo et al., 2017; Reusser and Zehe, 2011; Throop et al., 2012) and trail cameras with marked stakes (Bongio et al., 2019; Dickerson-Lange et al., 2017; Farinotti et al., 2010; Fortin et al., 2015). These options are relatively low-cost ($250 CAD [trail camera] to $700 CAD [10 iButtons] per stake) but have clear disadvantages. For example, iButton temperature loggers can have a low precision ($\pm0.5°C$) and sampling frequency (4-h sampling rate for less than a year of data) (Lewkowicz, 2008), experience frequent clock slippage and require specific modifications due to imperfect waterproofing. Trail camera

setups often require extensive manual processing, depend on weather conditions (interpretable images, camera battery life) and do not allow determination of other snow characteristics beyond snow heights (Farinotti et al., 2010; Garvelmann et al., 2013).

In this study, we present results from a novel low-cost technique for snow depth estimation that can be efficiently applied at remote field sites. The method we propose alleviates some of the challenges associated with other low-cost methods

while offering a relatively unambitious method of estimating snow characteristics in natural conditions. Building on the practice of using temperature loggers (Danby and Hik, 2007; Lewkowicz, 2008), we propose the snow characterization with light and temperature (SCLT) technique which uses vertically arranged dual light & temperature data loggers together to produce reliable estimates of snow characteristics with minimal analysis across ecotones. We tested the SCLT method for one year at six field sites located in forested and shrub-tundra locations in Subarctic and Arctic Labrador, north-eastern Canada.

Our results show sufficient promise that we believe there is significant benefit to sharing first results with the broader northern science community. Adoption of this method will facilitate a more prolific network of snow measurements in real-world conditions and will inform modelling and climate change adaptation measures while enhancing core understanding of cryospheric processes.



## 2 Study Area

The snow characterization with light and temperature (SCLT) method was tested at six field sites located in Subarctic and Arctic Labrador (northeast Canada). Field sites were within regions governed or managed by the Nunatsiavut Government, NunatuKavut Community Council and/or Innu Nation. The overall region has a strong coastal-continental gradient in air temperature, with higher snowfall amounts and colder temperatures than similar western Canadian latitudes due to the Labrador Current (Banfield and Jacobs, 1998; Brown et al., 2012; Maxwell, 1981; Way et al., 2017). Mean annual air

temperature ranges from around -8°C (Torngat Mountains Ecodistrict) to 2°C (L'Anse Amour Ecodistrict) and regional total precipitation ranges from 546 mm (Cape Chidley Ecodistrict) to 1248 mm (Mealy Mountain Ecodistrict) (Riley et al., 2013). On average, regional snow and ice cover is present from November to May (Brown et al., 2012); however, snow cover duration has rapidly declined in northern Labrador and climate models predict further reductions in snow cover duration in the future (Barrette et al., 2020; Brown et al., 2012). The six SCLT field testing sites (Table 1) cover a latitudinal range of 52.7°N to

58.5°N and are mostly located in forested ecodistricts (high Boreal forest, low Subarctic forest and mid Subarctic forest) where the dominant vegetation types are Black Spruce, White Spruce, Balsam Fir and Eastern Larch (Roberts et al. 2006; Riley et al., 2013) (Fig. 1; Table 1). One site (BaseSnow) is located in low-Arctic shrub-tundra (Torngat Mountains Ecodistrict) where dominant upright shrub species are Alder and Dwarf birch (Riley et al., 2013). The forested sites (Amet11, Amet12, Amet17, Amet28 and Amet19) are at a lower latitude and receive at minimum 7.6 hours of daylight while the higher latitude shrub-

tundra site (BaseSnow) at minimum 6.3 hours of daylight (Bird and Hulstrom, 1981).



**Table 1: Site specifications for the six SCLT sites including site name, latitude, longitude, elevation, ecotype and SCLT data collection period.**

| Site ID | Full site name | Latitude (°N) | Longitude (°E) | Elevation (m) | Vegetation ecotype | SCLT data collection period |
|---|---|---|---|---|---|---|
| **Amet11** | Mealy South Lower | 52.83 | -60.10 | 265 | Taiga forest | 2018-09-13 to 2019-07-24 |
| **Amet12** | Mealy South Upper | 52.79 | -60.03 | 467 | Taiga forest | 2018-09-13 to 2019-07-24 |
| **Amet17** | Goose Bay Upper | 53.30 | -60.54 | 271 | Boreal forest | 2018-10-14 to 2019-08-05 |
| **Amet28** | Aliant Tower Lower | 53.09 | -61.80 | 390 | Taiga forest | 2018-09-03 to 2019-08-12 |
| **Amet29** | Aliant Tower Upper | 53.11 | -61.80 | 526 | Taiga forest | 2018-09-03 to 2019-08-12 |
| **BaseSnow** | Torngat Basecamp | 58.45 | -62.80 | 3 | Shrub tundra | 2018-08-07 to 2019-08-19 |





**Figure 1: Geographic distribution of light and temperature snow stake sites (left) with detailed topographic depictions of each site (right).**

## 3 Methods

### 3.1 Theoretical Approach

The snow characterization with light and temperature (SCLT) method is based on prior research demonstrating that light transmission is inhibited by snow cover, and that overlying snow layer characteristics impact the magnitude and rate of light transmission through the snowpack (Fig. 2) (Libois et al., 2013; Perovich, 2007). The SCLT method is an evolution of a low-cost method, first described by Danby and Hik (2007) and Lewkowicz (2008), that uses vertically arranged temperature measurements and diurnal temperature fluctuations to estimate the date of snow cover at a given height (Lewkowicz, 2008).

SCLT uses simultaneous measurements of light intensity and temperature together to characterize snowpack characteristics.

**Figure 2: Conceptual diagram of the snow characterization with light and temperature (SCLT) method as implemented in this study. It is hypothesized that increases in snow depth will lead to sudden drops in light intensity measured by data loggers due to scattering and reflection in the snowpack (Perovich, 2007). A snow-covered logger is assumed to have mean values which are lower than ambient light intensity while temperature is assumed to remain at or just below freezing. Increased snow depth is assumed to result in less light penetration and decreasing diurnal temperature variation at lower logger heights. Impacts of snow aging and density variations are expected to impact these processes but are not explored in the present analysis.**

### 3.2 Field Implementation of SCLT method

Wooden stakes (1.8 m) were outfitted with vertically arranged HOBO MX2202 Pendant Wireless Temperature/Light Data Loggers (Onset Computer Corporation, 2020) anchored to 1.0 m metal poles driven into the ground (Table S1). Loggers were positioned at heights of 10, 20, 30, 40, 50, 60, 80, 100, 120 and 160 cm above the ground surface and thus characterize near-surface snow layers at a higher resolution than upper layers (Fig. 2). Visible light intensity and temperature was recorded at intervals of 2 hours (even intervals) and data was downloaded in the field via the HOBOmobile app (Onset, 2017). At each site, ground surface temperature, ground temperature (approximately 1 m depth) and air temperature were also collected





following Way and Lewkowicz (2018). Initial testing of the SCLT method covered the period of September 2018 to August 2019.

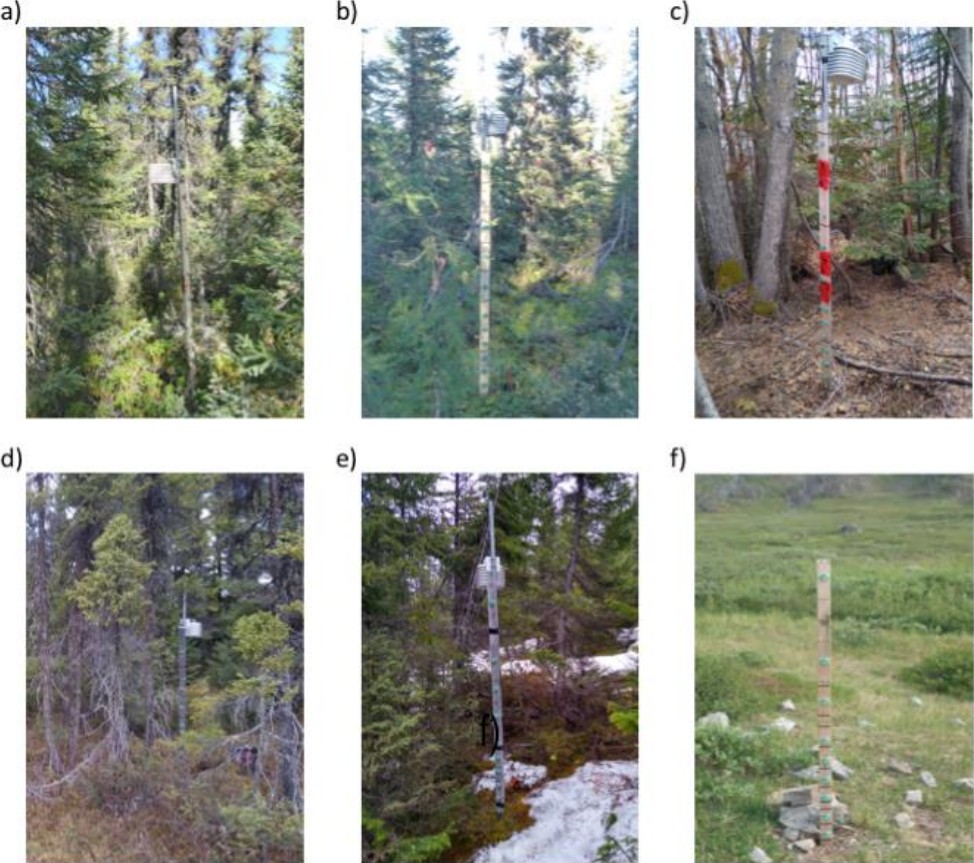

**Figure 3: Field photos of all SCLT measurement sites for 2018-2019. These include: (a) Amet11; (b) Amet12; (c) Amet17; (d) Amet28; (e) Amet29; and (f) BaseSnow.**

### 3.3 Data Processing and Analysis

We determined snow surface heights using SCLT using three unique but conceptually similar approaches. All analyses assume that snow cover at a given height occurs when daily maximum light intensity or daily temperature standard
deviation drops below an empirical threshold. The first approach applied changepoint analysis to raw light intensity measurements with the assumption that sudden changes in light intensity recorded at a logger are indicative of complete or partial snow coverage. The position of changepoint segments was determined using the Pruned Exact Linear Time (PELT) test method (asymptomatic penalty of 10%) which provides moderate sensitivity (Aminikhanghahi and Cook, 2017) and fast





processing time (Beaulieu et al., 2012; Wambui et al., 2015). A logger is deemed snow covered if a drop in light intensity

causes changepoint segments to fall below a threshold derived empirically.

Snow cover thresholds were defined as the minimum of the daily maximum light intensities during no-snow conditions at a data logger. No-snow conditions were considered days where the daily maximum temperature recorded at a given logger was above 0.5°C. This approach resulted in thresholds and ranges of daily maximum light intensities that varied from logger-to-logger (Fig. 4; Fig. S1). Application of changepoint analysis with the empirical thresholds enabled detection

of stepwise increases (or decreases) in snow surface heights relative to a logger's position (Fig. 5). Estimated snow depth was floored to the closest logger height which, when using raw data, resulted in uncertainties of $\pm$ 10 cm at lower positions and up to $\pm$ 40 cm for the top position.

The second approach applied to SCLT data uses similar logic as the first method but takes advantage of the high correlation between loggers at different heights through interpolation (Table S2). Daily maximum light intensity data was

interpolated using a modified thin plate spline interpolation designed for spatial processes from the fields R package (Nychka et al., 2017). Mean Absolute Error (MAE) of daily maximum interpolations ranged from 0.089 - 0.398 lux (logarithmic) for light and 0.099 - 2.01°C for temperature (Table S2). Snow cover was estimated from interpolated SCLT data with two different techniques: (1) standard changepoint analysis (PELT method, asymptomatic penalty of 10%) using the mean threshold using pooled data for all loggers at a given stake; and (2) using the minimum, mean and maximum of the empirical snow cover

thresholds from all loggers across a stake (contour method) (Fig. S1).

A third approach based entirely on temperature (Fig. S2) was used for comparison with the light intensity-based methods presented above. Estimation of snow depth with only temperature data is widespread in the ecological and permafrost literature and relies on measuring attenuation of diurnal variability in the snowpack (Danby and Hik, 2007; Lewkowicz, 2008). We apply changepoint analysis (PELT method, asymptomatic penalty of 10%) to daily temperature standard deviations

measured at each logger using the minimum standard deviation measured during no snow conditions (Tmax > 0.5° C) for each





height as an empirical threshold. A second condition was added where minimum temperature on a given day must be less than

or equal to 0.5° C for snow cover to be present.

**Figure 4: Violin plot (rotated kernel density) showing the probability density and distribution of daily maximum light intensities**
**(logarithmic scale) when the daily maximum temperature is above 0.5° C at: (a) Amet11, (b) Amet12, (c) Amet17, (d) Amet 28, (e) Amet29 and (f) BaseSnow. Minimum values were used as the individual logger thresholds for the changepoint analysis and pooled thresholds were used for the range of thresholds used in the interpolated analysis.**


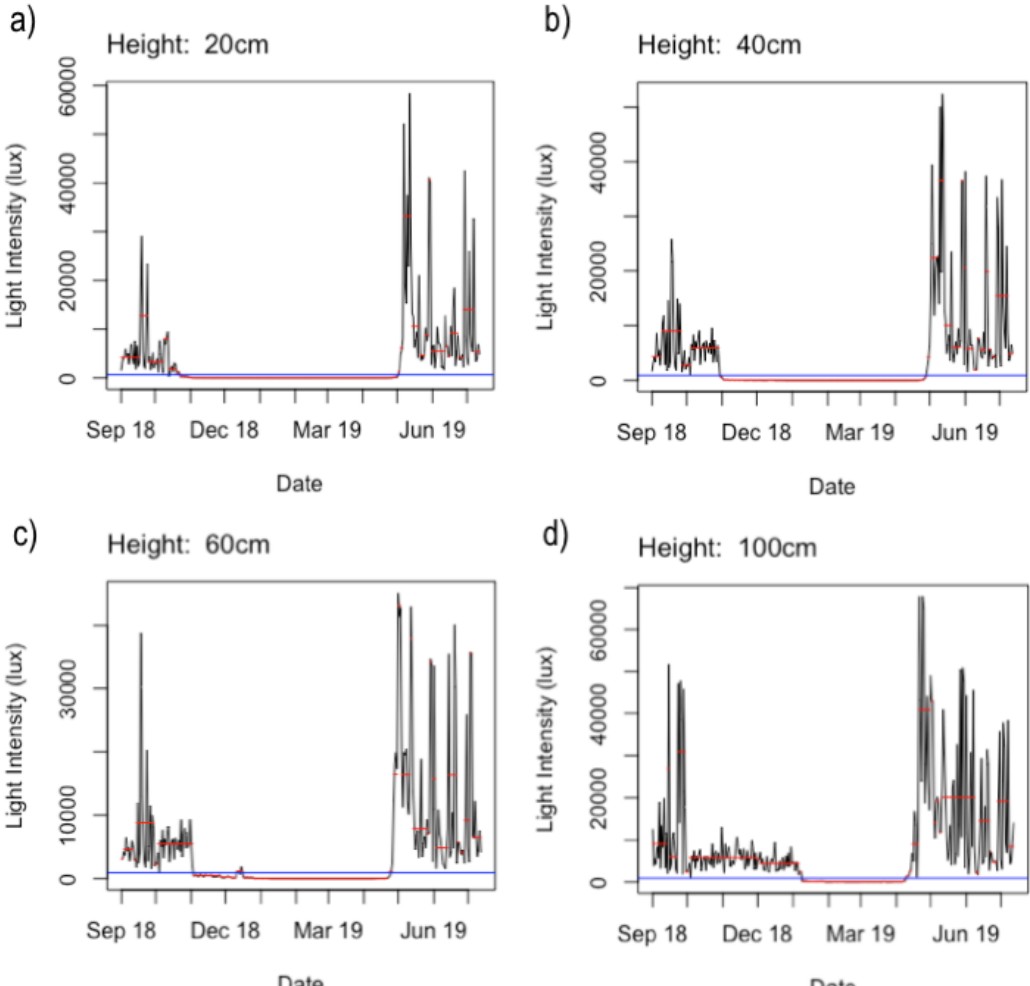

**Figure 5: Changepoint analysis applied to (a) 20, (b) 40, (c) 60 and (d) 100 cm height loggers along Amet11. The red line shows**
**changepoint segment means and the blue line shows the no-snow light intensity threshold for each logger. Snow cover occurs at a**
**given logger when the changepoint segment drops below the no-snow threshold.**

## 4 Results

### 4.1 Estimating snow depth using lux measurements

We used the SCLT method to estimate snow depth through the winter for 2018-2019 at six remote sites across
Labrador. The first analysis method derives the snow depth using a changepoint analysis of the raw daily aggregates and the
second uses interpolated light intensity data. A third method is entirely based on temperature and is presented for a comparison
to data analysis methods used in prior studies.





## 4.2 Changepoint analysis with raw light intensity measurements

At forested sites (Amet11, Amet12, Amet17, Amet28, Amet29), snow accumulated stepwise beginning in mid-
October with a maximum depth reached between March and April followed by rapid snow melt in early-to-mid May (Fig. 6).
At the shrub-tundra site (BaseSnow), snow cover was generally thin over much of the winter with smaller periods of
accumulation in the late-fall and early-winter. At BaseSnow, maximum snow thickness was reached in mid-March to mid-
April and a complete melt occurred by early-May. Across all sites the snow cover duration ranged from 174 days (BaseSnow)
to 229 days (Amet12) with an average duration of 215 days (Table 2). Mean January snow depth was also lowest at BaseSnow
(~11 cm) and highest at Amet12 (~103 cm). In 2018-2019, all SCLT sites except for BaseSnow had a snowpack taller than the
uppermost data logger (160 cm; 120 cm at Amet11 due to a logger failure) for anywhere between 8 days (Amet28) to 84 days
(Amet11) (Table 2).

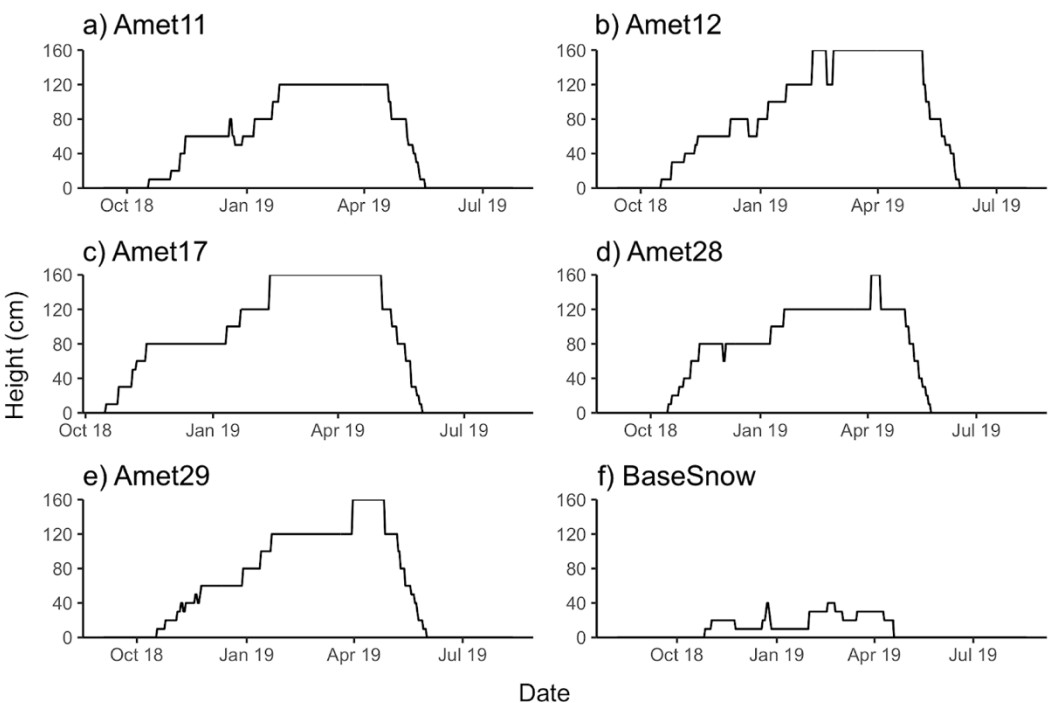

**Figure 6: Snow depth over 2018-2019 derived using changepoint analysis of raw lux values from loggers at each SCLT measurement
site including: (a) Amet11, (b) Amet12, (c) Amet17, (d) Amet28, (e) Amet29 and (f) BaseSnow. Top logger positions ranged from 120
cm (Amet11 and BaseSnow) to 160 cm (Amet12, Amet17, Amet28, Amet29) and cannot detect snow depths above this height.**

**Table 2: Snow cover duration, maximum snow depth, duration at maximum depth and mean January snow depth for each SCLT
site for 2018-2019 using the changepoint method with raw lux values.**

| Site | Snow cover duration | Maximum snow depth | Duration at max depth | Mean January snow depth |
| --- | --- | --- | --- | --- |





| | | | | |
|---|---|---|---|---|
| Amet11 | 212 days | > 120 cm | 84 days | 87.1 cm |
| Amet12 | 229 days | > 160 cm | 80 days | 103.2 cm |
| Amet17 | 228 days | > 160 cm | 81 days | 100.6 cm |
| Amet28 | 220 days | > 160 cm | 8 days | 101.3 cm |
| Amet29 | 226 days | > 160 cm | 27 days | 98.7 cm |
| BaseSnow | 174 days | 40 cm | 9 days | 10.6 cm |


## 4.3 Snow depth estimation with interpolated light intensity measurements

Light intensity was interpolated along each stake and two analysis techniques were applied to the interpolated data (Fig. 7). The first, which used changepoint analysis, showed small increases in snow accumulation from late-October to late-January for Amet11, Amet12 and Amet 17 with snow cover above the top logger (greater than 120cm for Amet11 and 160cm

for Amet12 and Amet17) until spring snowmelt in late-April to early-May. With the interpolated changepoint method, Amet28 accumulated snow until April when it reached a maximum snow depth of 133 cm on March 21, 2019 and melted from late April until mid-May. At Amet29 snow depth exceeded the top logger (160cm) from mid-to-late April and melted throughout May (Fig. 7). BaseSnow showed a thinner snow cover with short periods of accumulation in the late-fall (November), late-December and February with a maximum snow depth of 31 cm in late-January. The interpolated changepoint analysis resulted

in snow cover durations ranging from 177 days (BaseSnow) to 234 days (Amet12) and mean January snow depth ranging from 17 cm (BaseSnow) to 120 cm (Amet17).

The second approach applied to interpolated data used the minimum, mean and maximum stake-wide pooled thresholds to produce a range of contours showing potential snow depths for each day. The SCLT snow depth using mean thresholds showed a similar pattern to the changepoint analysis described above with accumulation from late-October to late-

January, with the notable exception that snow cover at Amet28 exceeded the top logger with this method (Fig. 7). BaseSnow showed dispersed accumulations between the late-fall and early-spring with rapid melt occurring in mid-April and a maximum snow depth of 43 cm on December 23, 2018. Snow cover duration ranged from 178 days (BaseSnow) to 200 days (Amet17) and mean January snow depth ranged from 23.0 cm (BaseSnow) to 120 cm (Amet17) (Fig. 7). Applying the contour approach to 2018-2019 winter SCLT data leads to mean time-varying snow depth uncertainty ranges from 3 ± 3 cm (Amet17) to 15 ± 6

cm (Amet28).

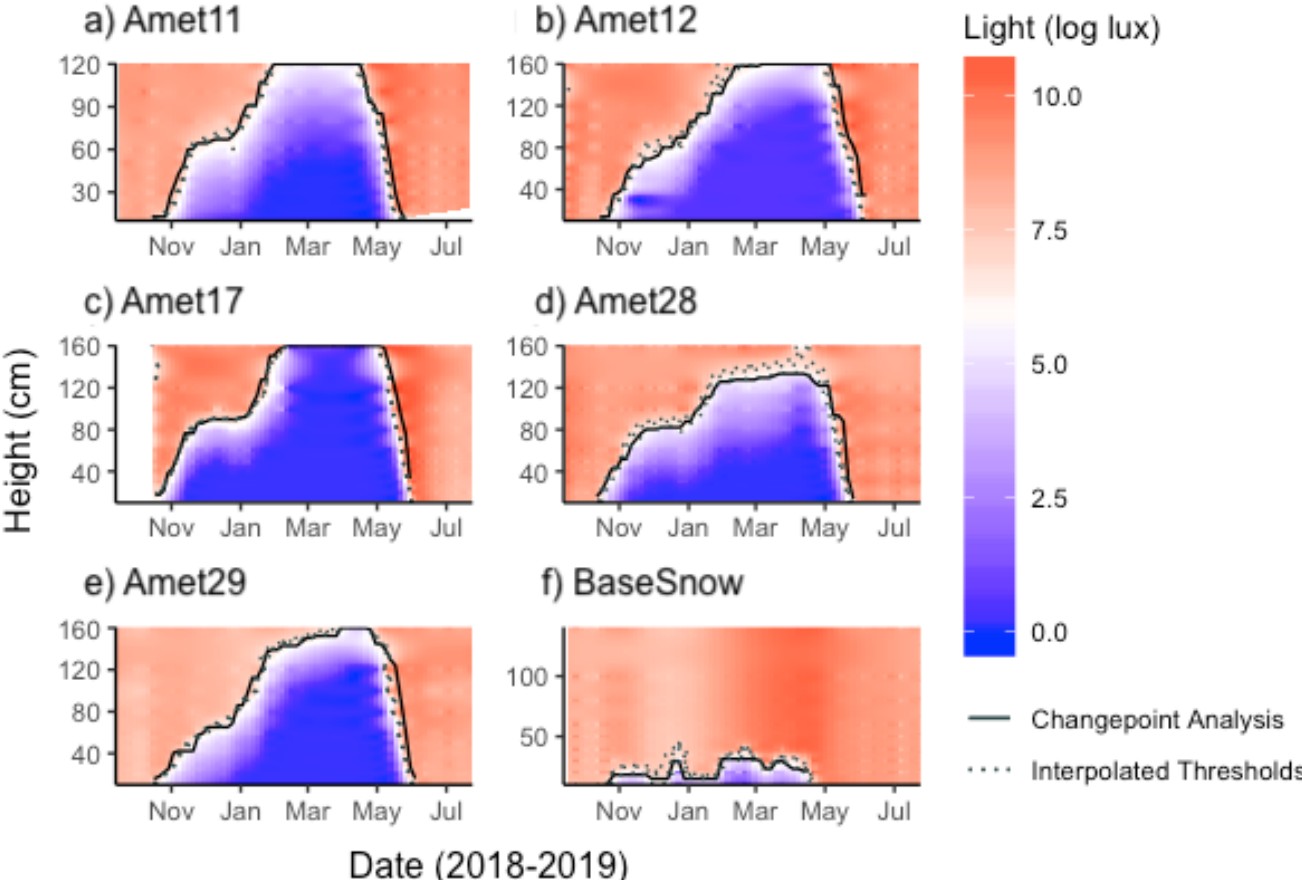

**Figure 7: Interpolated lux measurements presented as an x-y-z plot on a log-scale for each SCLT measurement site including: (a) Amet11, (b) Amet12, (c) Amet17, (d) Amet28, (e) Amet29 and (f) BaseSnow. Estimated snow depths are presented for changepoint analysis (black) and the mean of the no-snow thresholds (contour-method; dotted).**

## 4.4 Estimating snow depth using temperature measurements

Application of the temperature-based changepoint analysis resulted in forested stations (all Amets) showing snow accumulation starting in mid-to-late October but not until late-December at the shrub tundra site (BaseSnow). All temperature-based snow depth estimates showed a drop in snow depth in late-December (Fig. 8). Amet11 reached a maximum snow depth of 100 cm in February but periodically dropped to 50 cm throughout the winter with a rapid decline in late-April to early-May (Fig. 8). Amet12 and Amet17 exceeded the top logger in February but had sudden drops in snow depth throughout the winter into early-spring. Amet28 and Amet29 both accumulated snow gradually until early-April with peak snow depths of greater than 120 cm and 160 cm, respectively. Melt is inferred to have occurred at all SCLT sites excluding BaseSnow between late-April and late-May. At BaseSnow, spikes in snow cover up to 30 cm occurred in late-December and late-March to early-April. Excluding these peaks, snow cover at BaseSnow remained at 0 cm throughout much of the snow season (Fig. 8). With the





univariate temperature analysis, snow cover duration ranged from 104 days (BaseSnow) to 227 days (Amet12 and Amet17) and mean January snow depth ranged from 0 cm (BaseSnow) to 101 cm (Amet12).

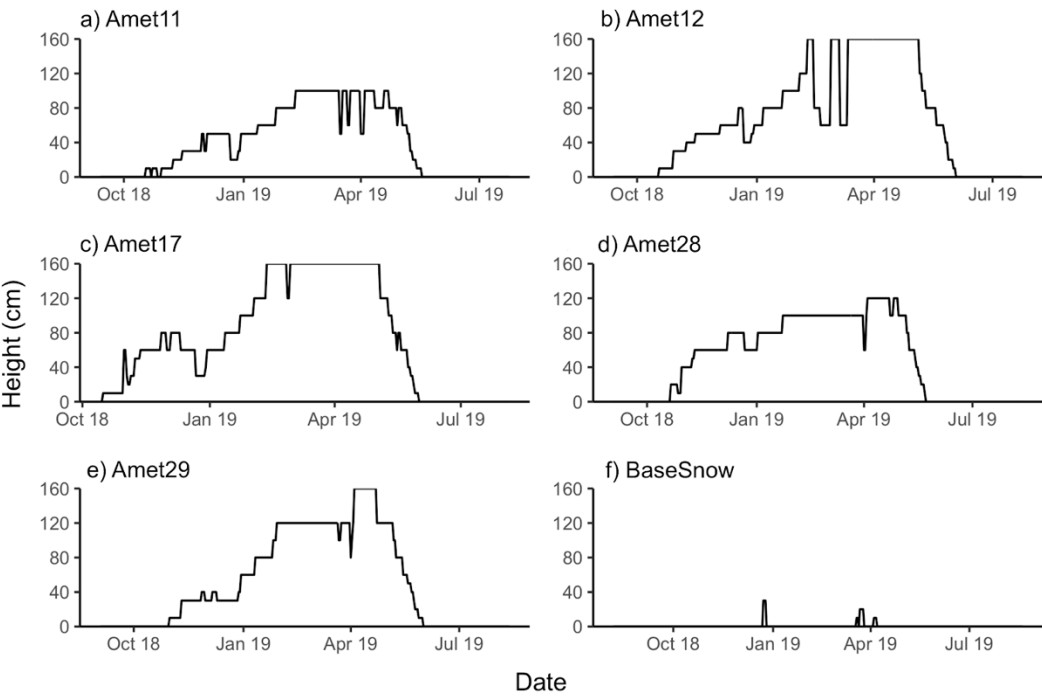


**Figure 8: Snow depth over winter 2018-2019 derived from changepoint analysis applied to standard deviations of daily temperature for each SCLT measurement site including: (a) Amet11, (b) Amet12, (c) Amet17, (d) Amet28, (e) Amet29 and (f) BaseSnow. Top logger positions for SCLT sites ranged from 120 cm (Amet11 and BaseSnow) to 160 cm (Amet12, Amet17, Amet28, Amet29) and cannot detect snow depths above this height.**


### 4.5 Comparison of SCLT lux methods

Raw and interpolated light intensity-based methods showed similar periods of snow onset with gradual snow accumulation from October to May for the Amet sites but the raw changepoint analysis resulted in a shorter duration of snow cover compared to the interpolated data at all sites (Fig. 7). Generally, the raw changepoint method showed larger single day-

increases in estimated snow depth, while the same method applied to interpolated data resulted in smaller, more frequent accumulations. Application of the contour method (using minimum, mean and maximum thresholds) resulted in smooth periods of accumulation and transport or melt but were mostly similar to the changepoint-based estimates (Fig. 7). Changepoint analysis and contours using interpolated data resulted in similar mean January snow depths for all stations with a mean difference of $3 \pm 2$ cm (Table 3). The mean January snow depth was significantly lower using the changepoint method on the

raw data at all stations, with differences ranging from 10.2 cm (Amet28) to 18.4 cm (Amet17) (Table 3).



Comparison of a forested (Amet12) and shrub-tundra site (BaseSnow) showed earlier snowmelt with the raw changepoint analysis at the former site but no clear differences in melt at the latter site (Fig. 9). The raw changepoint method also showed a period of snow removal or melt in the early-to-mid winter at the forested site though this was not evident in the interpolated data (Fig. 9). All three light-based methods showed a consistently low snowpack at the shrub-tundra site

(BaseSnow) with greater overall variability in the raw changepoint analysis (Fig. 9).

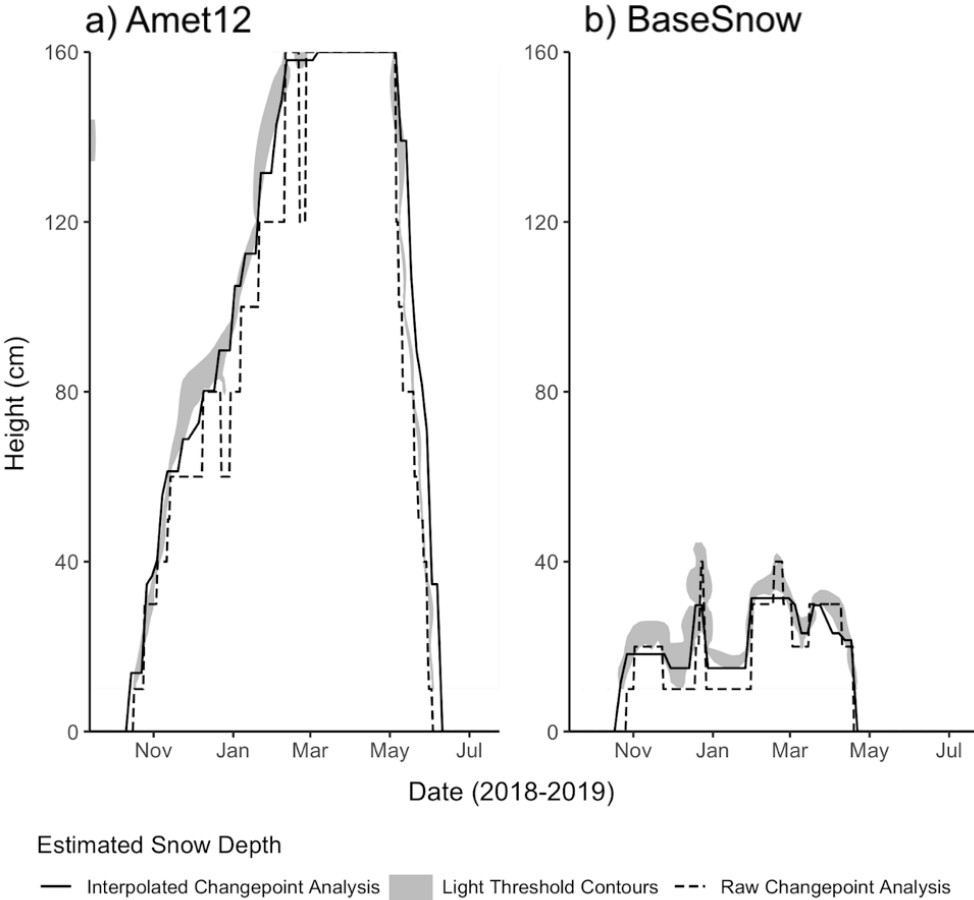

**Figure 9: Comparison of snow depths derived with light-based methods for: (a) a forested site (Amet12) and (b) a shrub-tundra site**
**(BaseSnow). Snow depth estimates are provided for raw changepoint analysis (dashed lines), interpolated changepoint analysis**
**(black line) and interpolated contours using minimum and maximum snow cover thresholds (grey shading).**

## 4.6 Comparison of light and univariate temperature methods

Estimated snow depth using temperature showed large drops in the late-Fall and mid-Winter at most sites that were not evident in the light intensity-based methods. Temperature-based snow depths consistently produced shorter snow durations





and less snow accumulation at all sites (Fig. 8; Table 3). For the forested sites (Amets), the differences in mean January snow depth between the temperature changepoint and the raw SCLT changepoint ranged from 2 cm (Amet12) to 22 cm (Amet17) (Table 3) though an even greater difference was found when comparing interpolated data (mean difference of 27 ± 11 cm). At BaseSnow (shrub-tundra), the temperature method estimated a snow depth of 0 cm in January while the light-based methods estimated mean snow depths between 10 cm and 23 cm (Table 3).


**Table 3:** Mean January snow depth for all six stations using all methods.

| Field site | Raw light changepoint | Interpolated light changepoint | Interpolated light threshold contours (mean) | Raw temperature changepoint |
|---|---|---|---|---|
| Amet11 | 87.1 cm | 100.7 cm | 98.2 cm | 69.0 cm |
| Amet12 | 103.2 cm | 117.8 cm | 120.7 cm | 101.2 cm |
| Amet17 | 100.6 cm | 120.1 cm | 119.0 cm | 78.7 cm |
| Amet28 | 101.3 cm | 107.6 cm | 111.5 cm | 96.8 cm |
| Amet29 | 98.7 cm | 115.2 cm | 114.0 cm | 81.9 cm |
| BaseSnow | 10.6 cm | 17.3 cm | 23.0 cm | 0 cm |

Temporal variability in snow depths was examined using Pearson correlation coefficients calculated across sites and methods between December and January (avoiding snow depths exceeding maximum logger heights). Amongst the four

methods examined, snow depths derived using light-based methods were highly correlated with one another (r = 0.7 to r = 0.98) but were much less correlated with the temperature-based snow depths (Fig. S3). Raw changepoint analysis using light provided the highest mean correlation with the temperature-based snow depths across sites (r = 0.85). Overall, cross-method correlations were highest for Amet29 and lowest for BaseSnow reflecting the highly variable snow conditions at the latter site (Fig. S4).

**5 Discussion**

**5.1 Evaluation of SCLT performance**

Evaluation of the snow characterization with light and temperature (SCLT) method in Subarctic and Arctic Labrador over winter 2018-2019 showed that the technique can reliably and consistently determine snow depth in both forested and shrub-tundra environments. The raw changepoint requires minimal processing time and is easiest to implement, but by ignoring

the inter-associations between measurements at different heights it will inherently floor snow depth to the closest logger leading to larger errors than with interpolated data. Interpolation of SCLT data was also able to compensate for logger failures,



particularly post-snow coverage, by using the high correlation between loggers within the snowpack to estimate missing data (SI Table 2). The univariate temperature analysis applied to our sites underperformed relative to the light-based methods with the divergence between approaches most evident at the shrub-tundra site (BaseSnow) (Fig. S4). The snowpack at this site was

inferred to be dense due to wind packing and thus would experience greater diurnal temperature variability because of a higher thermal conductivity compared to a forest site (Domine et al., 2016; Sturm et al. 1999). The high light intensities outside of the snowpack induced by the albedo effect provided a fairly unambiguous contrast with the lower light intensities within the snowpack (Fig. 7), allowing for depth determination of a snowpack that is typically difficult to characterize (Domine et al., 2019).

As elucidated by Sturm et al. (2001), snow cover is sensitive to local micro-climate, vegetation cover and topography. These variables are not broadly represented in current weather monitoring infrastructure deployed near urban centres or airports (Goodison, 2006). The lack of weather stations recording snow depth adjacent to our field sites makes it difficult to validate results from most SCLT sites. However, Amet17 is located approximately 5 km from Goose Bay Airport which has a weather station measuring snow depth though this site is found in an open clearing and at a site that is 200 m lower than Amet17

(Environment and Climate Change Canada, 2020). Comparing the two 2018-2019 snow depths from both sites shows high general agreement (r = 0.98 for daily snow depths from December to January [n=112]) but Amet17 showed a longer overall snow season and a significantly later snow melt than at Goose Bay Airport (Fig. 10). This difference is not unexpected as Brown et al (2003) showed a thicker peak snow depth and longer snow duration at forested versus open snow course sites (currently inactive) near Goose Bay. Later snow melt at Amet17 can also be inferred from a site visit to Amet17 in 2020

(March 25) which showed a significantly thicker snowpack at Amet17 (95 ± 5 cm; Fig. S5) than contemporaneously measured at Goose Bay Airport (52 cm) (Environment and Climate Change Canada, 2020).

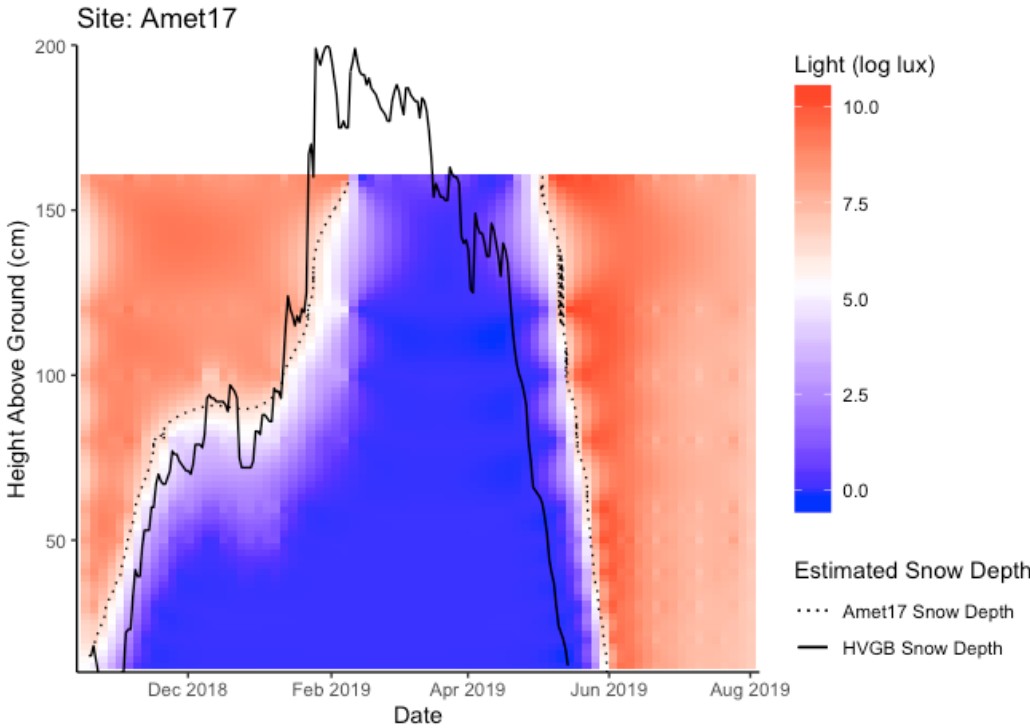

**Figure 10: Estimated snow depth at Amet17 site (black dotted line) using interpolated SCLT data overlaid with snow depth measured**
**for winter 2018-2019 at Goose Bay Airport (black solid line).**

## 5.2 Limitations and Opportunities

The results in this study have provided a direct workflow for estimating snow depth from SCLT data though the
proposed method will require further optimization and refinement. For example, our analysis did not directly evaluate the
impacts of latitude, canopy cover, logger configuration and ground condition on SCLT results. Each of these factors and their
corresponding influence on light transmission under snow and no-snow conditions makes the universal application of particular
light thresholds unlikely. The specific sensor arrangement of SCLT stakes may also require refinement and customization for
indices studied. Winter 2018-2019 far exceeded normal snow depths in coastal Labrador (Figure S6), resulting in data gaps
mid-winter. The configuration in this study was designed for investigations of ground thermal impacts of snow cover in
discontinuous permafrost in Labrador which typically are largest when snow cover is shallower than 100 cm (Way and
Lewkowicz, 2018). For hydrological applications, uniform sensor arrangement at a given interval (e.g. 5-10 cm) may be
preferable. Field visits to sites also suggest that maintaining a consistent measurement height may be challenging in areas with
significant frost heave from year-to-year therefore alternative anchoring may be needed for examining changes at a site over
multi-year periods. The widespread applicability of SCLT will depend on further testing at high latitudes where the lack of



light availability during December and January may limit its utility during portions of the winter. However, this concern may be limited to the short periods of complete darkness as we observed substantial light reflection from high albedo tundra snow cover at our highest latitude site (BaseSnow) even in December. Exploring the potential utility of combining light intensity and temperature together with more advanced predictive modelling may further mitigate this concern. We would also recommend that a specific sensor arrangement pointing south or towards the most open portion of the canopy could be adopted

to enhance light intensity contrasts at low sun angles.

Overall, the SCLT method was found to provide robust and cost-efficient snow depth estimation in regions that are not suitable for outfitting with full weather stations. We unambiguously show that light intensity is a clearer metric for estimating daily snow depth than temperature-only methods. Further analysis combining the light intensity measurements with temperature within the snowpack will allow for a more robust snowpack characterization than available through the use of

time lapse photography-based methods. The dual measurements collected by the SCLT technique coupled with ground temperature measurements will also enable simplified characterizations of temperature gradients within the snowpack and at depth as a coupled system (Fig. 11). Further studies should explore how SCLT can be applied to better understand other snowpack characteristics including density, grain size and effective thermal conductivity.


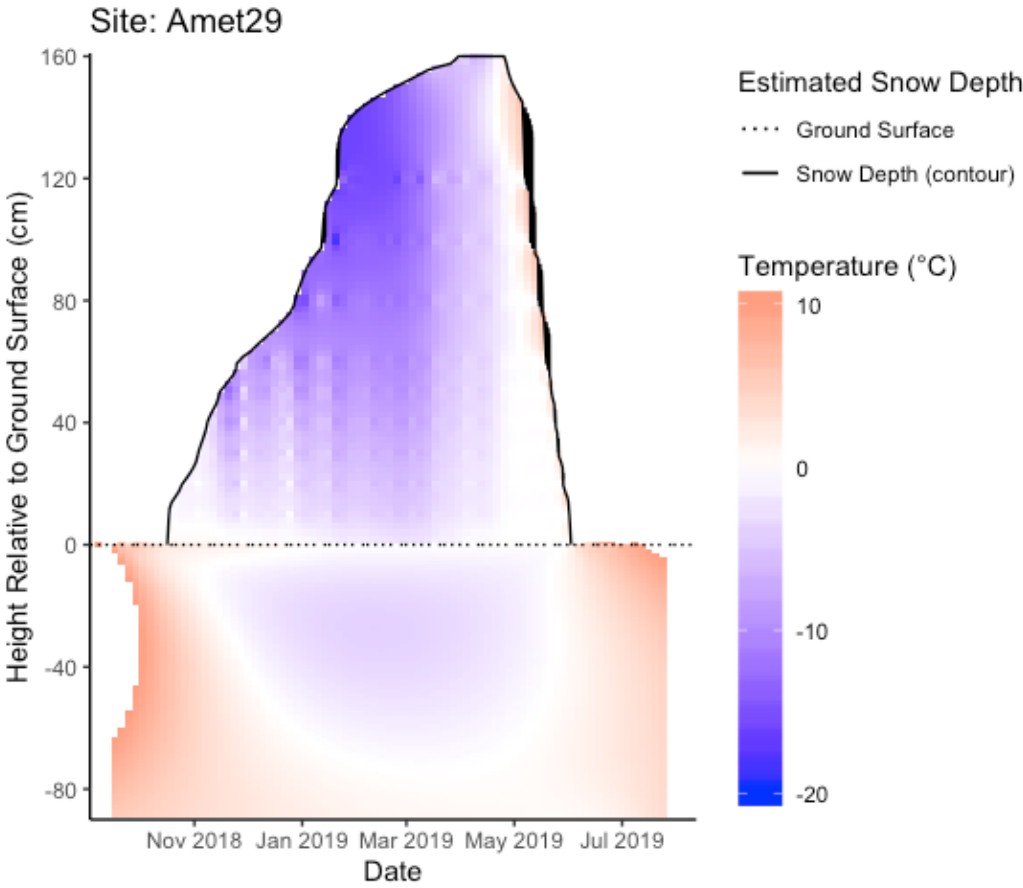

**Figure 11: X-Y-Z plot showing interpolated temperatures for Amet29 within the snowpack and the underlying soil (maximum depth: 85 cm). Snowpack height is estimated using the interpolated light threshold contour (mean) (black line) and ground temperatures were recorded at 5 cm and 85 cm depth with a Hobo V2 Pro data logger.**

## 330  6 Conclusion

Improved monitoring and characterization of a changing snowscape is imperative to conservation, planning and climate adaptation in across the globe but particularly in Subarctic and Arctic regions. Snow characterization under natural environments is currently lacking in most northern environments with measurement stations mostly in open areas near airports or communities making snow studies outside of these regions dependent on snow courses and remote sensing (Brown et al., 335  2003; Goodison, 2006; Pulliainen et al., 2020). In this study, we introduce a novel method (SCLT) for characterizing snow conditions in remote northern environments that uses a combination of vertically arranged light and temperature loggers. We present three different methods for analyzing SCLT data, including a temperature-only approach for comparison with prior studies. Our results broadly show that raw and interpolated SCLT data can be used to efficiently characterize snow depth over full snow seasons at sites that varied considerably in ecotype and inferred snow characteristics. All SCLT-based snow



estimation techniques provided clear advantages over the temperature-only approach with the latter performing particularly

poorly where snow density was inferred to be higher (shrub-tundra).

The development of the SCLT method as a cost-effective measurement technique aims to help fill knowledge gaps

in snow-vegetation interactions and to facilitate a wider snow monitoring network in remote areas under natural conditions.

The method requires further research and refinement; however, these preliminary results are sufficiently promising that

deployment of SCLT across northern research basins for testing purposes may be desirable. Applying this new method will

improve our understanding of the changing cryosphere, local hydrology and climate change impacts on ecosystems and

biodiversity. Further elucidation of snow-vegetation-permafrost interactions will also aid community development, local travel

safety and cultural practices.

**Code and Data Availability**

The SCLT data contributes to a larger dataset presented by Way and Lewkowicz (2018) that will be made available through

Nordicana D. The R v3.6.0 or RStudio v1.2.1335 code for: (a) inputing and preprocessing HOBO Pendant Light/Temperature

csv data (b) determining light thresholds and (c) snow depth evaluation through changepoint analysis and interpolation, are

available through the authors' ResearchGate repository at the doi links below. Additional code is available upon request.

    (a)   Tutton, R. and Way, R.: SCLT Data Pre-processing, , doi:10.13140/RG.2.2.17281.48483, 2019.

(b)   Tutton, R. and Way, R.: SCLT Threshold Determination, , doi:10.13140/RG.2.2.14093.15841, 2019.

    (c)   Tutton, R. and Way, R.: SCLT Snow Cover Determination (Changepoint), , doi:10.13140/RG.2.2.35064.67843, 2020.

**Author Contribution**

RGW established snow stake field sites as part of a wider permafrost monitoring network. RT and RGW collected and analysed

field data and reviewed results relative to local weather monitoring stations. RT and RGW drafted the final manuscript.

**Competing Interests**

The authors declare that they have no conflict of interest.

**Acknowledgements**

The authors would like to acknowledge that research activities were undertaken on land originally and contemporarily

occupied by the Innu and Inuit people of Labrador represented by the Innu Nation, Nunatsiavut Government and NunatuKavut

Community Council. We would like to thank the Nunatsiavut Government and the Western Newfoundland and Labrador Field

Unit of Parks Canada for direct logistical and research support for activities at the Torngat Mountains Basecamp and Research



Station. This research benefited from conversations and/or logistical support from Caitlin Lapalme, Dr. Darroch Whitaker, Rodd Laing, Dr. Antoni Lewkowicz, Ross Brown, Dr. Philip Bonnaventure, Dr. Luise Hermanutz, Dr. Andrew Trant, Dr. Sharon Smith, Charlene Kippenhuck, George Russell Jr., Yifeng Wang and Taylor Larking. This research was financially and/or logistically supported by Queen's University, the W. Garfield Weston Foundation, the Labrador Institute of Memorial University of Newfoundland, the Nunatsiavut Government and ArcticNet Inc. Network of Centre of Excellence.

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
