# Peer review of "A low-cost method for monitoring snow characteristics at remote field sites"

_The Cryosphere, 2020_

## Referee Comment (RC1) · Michael Prior-Jones (Referee) · 19 Aug 2020

General comments

This paper fulfils all the criteria for publication and I would be happy to see it published given a few minor changes. The background, prior work and fieldwork methodology are all explained clearly. The main issue that I would like to see addressed is that the authors lead the reader to expect the presentation of a method that combines the measurements of both light intensity and temperature to produce a snow depth measurement. The abstract promises "a new method termed snow characterization with light and temperature", but while both parameters are measured, they are then analysed separately and the results from each approach compared. The temperature

sensor is used in the light intensity method but only to determine snow-free days for the baseline calibration. The authors do acknowledge that they intend to develop a technique that incorporates both kinds of measurements in future work, but it would be good to clarify this up front.

Specific comments

Introduction, line 30: "Unlike its liquid counterpart, snow is ..." – I presume by this you mean by comparison with rain? If that's the case, I think it would be better if you revised the sentence and explicitly compared snowfall with rainfall.

Introduction, line 35. I would suggest also making reference to the fact that snow does not lie evenly because of the effects of topography and wind (snowdrifts, sastrugi, etc), so extrapolating local or regional levels of snow cover from a single point measurement is very prone to errors. The development of a low-cost technique potentially allows multiple instruments to be deployed within a region of interest to get a more representative measurement of snow cover.

Data processing and analysis, line 134. On my first read-through, I struggled to understand whether the interpolation discussed here between loggers on a single stake, or was interpolation in time between logger readings. The explanation (it's between loggers on a stake) doesn't come until some time later at the start of section 4.3, so it is worth a clarification here.

Results, line 183: "The first, which used changepoint analysis, showed small increases in snow accumulation from late-October to late-January..." – I presume that these "small increases" were relative to the non-interpolated method, but it would be good to have this clearly stated.

Congratulations on your paper, it's a very creditable piece of work!

Dr Mike Prior-Jones, Cardiff University, UK.

---

## Referee Comment (RC2) · Anonymous Referee #2 · 1 Sep 2020

Snow depth is a poorly observed variable: measurement networks are sparse (especially across the subarctic and Arctic) the measurements are prone to uncertainty, and often fail to capture the prevailing landscape-scale variability. Limited tools are available to address these weaknesses, so advancements in low-cost, easy to deploy instrumentation is of high interest to the snow community.

This manuscript provides an overview of a new low-cost approach to acquiring snow depth estimates using vertical profiles of light and temperature measurements. The paper is clearly written and provides a careful intercomparison of different processing techniques to derive snow depth from the profile measurements.

While I have only a small number of suggestions on the manuscript, my main concern, as outlined in my first comment below, is with the experiment design.

[Figure]

1. Because no independent time series of snow depth measurements were acquired coincident to the profile measurements, it is not possible to know what the 'true' snow depth was. This means the various approaches to estimating snow depth can be compared, but not assessed. Only in the case of Goose Bay can the SCLT-derived snow depth time series be shown alongside an independent measurement, and as rightly pointed out in the text, this comparison serves to highlight the differences between snow measurements made in open environments (airports) versus in the forest. Ideally, an SR-50 or another independent sensor would have operated alongside the SCLT profile, although I understand the cost of such a deployment could be prohibitive. To mitigate the lack of assessment and highlight the inter-comparison, it would be useful to see the multiple snow depth time series produced at each location from the various techniques on a single plot (individual panel for each site). This would illustrate the range/agreement in snow depth through time based on the analysis methodology, which can now only be inferred by flipping between figures, and looking at the correlation results in Figure S3. I suggest adding this new figure, and a brief discussion of it, to Section 5.

2. Line 262-268: consider moving Figure S3 out of the supplemental material to include it in this paragraph. I think these are worthwhile results to include in the manuscript.

3. This issue is acknowledged in the Discussion, but why not use consistent vertical spacing of 10 cm? In a relative sense, greater uncertainty with deeper snow is ok, but the current setup dictates that uncertainty will be greater when snow is deeper.

4. Line 122: can you provide a simple description of the PELT method? Not clear what is meant by 'asymptomatic penalty of 10%'.

5. Line 170: "all SCLT sites except for BaseSnow had a snowpack taller than the 170 uppermost data logger". Murphy's Law at work that there was an unusually deep snowpack during the season that you were evaluating this approach! Do you have a sense of how tall the profile needs to be? Is there any technical limitation to say, a 2 m

profile with sensors every 10 cm?

6. Cost effectiveness is a major driver of this work, but (unless I missed it) nowhere in the paper is the cost of the SCLT profiles stated. This information would be helpful! What is the cost sensitivity to the vertical resolution of the profile?

7. Sections 4.5 and 4.6 provide more detailed analysis of the light intensity methods. To improve the logical structure of the paper, I suggest shifting these up to follow Section 4.3, and shift down the temperature measurement approach reported in Section 4.4.

8. The reference list needs to be cleaned up. Some citations are missing (e.g. Archer, 1998) and details are missing from some references (journal titles, etc.). Review the sequence of figure numbers: figures jump from 10 to 13 to 15.

9. While the code and data availability are provided, what about schematics to the design of the probe? Will these be shared in some form so that others can follow your design if interested?

Editorial Line 30: change to 'snowfall is hard to catch, melts differentially once on the ground...'

Line 55: not clear what is meant by 'relatively unambitious method'...uncomplicated?

Line 60: suggest changing to 'broader snow science community'

Line 118: change to "We determined SCLT-derived snow surface heights using..."

Figure 4: minor point, but the y-axis range for BaseSnow is slightly different from the other sites.

---

## Author Response (AR1)

**AUTHOR RESPONSES TO EDITOR COMMENTARY ON TUTTON AND WAY MANUSCRIPT**

**[T&W response] We would like to thank Dr. Bagshaw for taking the time to provide helpful comments on our manuscript. We have responded to each comment below and have made the suggested changes to the final revised manuscript. A short summary of implemented changes is at the bottom of this response.**

*Please could you revise the first sentence in the abstract: it is now rather overlong and/or lacking commas*

**[T&W response] We agree with is comment and have changed the wording.**

*L14: no need for a hyphen in 'one-year'*

**[T&W response] We agree with this comment and changed hyphen.**

*A close up photo of the loggers mounted onto the stakes would be a great addition to Figure 3 – the general set up can be seen, but a close up of the sensors and the mounting would be beneficial to readers.*

**[T&W response] We agree with this comment and have added a photo with a close-up to Figure 3. Thanks again for this suggestion.**

**SUMMARY OF CHANGES TO MANUSCRIPT:**
**[1] Modification to abstract.**
**[2] Removal of hyphen.**
**[3] Addition of close-up photo in Figure 3.**

**AUTHOR RESPONSES TO REVIEWER 1 COMMENTARY ON TUTTON AND WAY MANUSCRIPT**

**[T&W response] We would like to thank Reviewer 1 (Dr. Prior-Jones) for taking the time to provide helpful comments on our manuscript. We have responded to each comment below and have made the suggested changes to the final revised manuscript. A short summary of implemented changes is at the bottom of this response.**

*The abstract promises "a new method termed snow characterization with light and temperature", but while both parameters are measured, they are then analysed separately and the results from each approach compared. The temperature sensor is used in the light intensity method but only to determine snow-free days for the baseline calibration. The authors do acknowledge that they intend to develop a technique that incorporates both kinds of measurements in future work, but it would be good to clarify this up front.*

**[T&W response] We agree and acknowledge that the methods outlined in this manuscript prioritize light measurements and have revised the introduction of this method to clarify that this is an evolving method and the temperature consideration requires further attention. See tracked changes to the revised manuscript.**

*Introduction, line 30: "Unlike its liquid counterpart, snow is . . ." – I presume by this you mean by comparison with rain? If that's the case, I think it would be better if you revised the sentence and explicitly compared snowfall with rainfall.*

**[T&W response] We agree with this comment and changed the wording to rainfall.**

*Introduction, line 35. I would suggest also making reference to the fact that snow does not lie evenly because of the effects of topography and wind (snowdrifts, sastrugi, etc), so extrapolating local or regional levels of snow cover from a single point measurement is very prone to errors. The development of a low-cost technique potentially allows multiple instruments to be deployed within a region of interest to get a more representative measurement of snow cover.*

**[T&W response] We agree with this comment and included the benefit of dispersed point measurements in the following paragraph with context.**

*Struggled to understand whether the interpolation discussed here between loggers on a single stake or was interpolation in time between logger readings. The explanation (it's between loggers on a stake) doesn't come until sometime later at the start of section 4.3, so it is worth a clarification here.*

**[T&W response] We agree with this comment and have clarified that the interpolation is along the x (time) and y (height along stake) throughout the text.**

*Results, line 183: "The first, which used changepoint analysis, showed small increases in snow accumulation from late-October to late-January..." – I presume that these "small increases" were relative to the non-interpolated method, but it would be good to have this clearly stated.*

**[T&W response] We agree that this phrasing is unclear and reworded it to simply describe the increasing snow cover at these stations using the changepoint method.**

**SUMMARY OF CHANGES TO MANUSCRIPT:**
**[1] Specifying the lack of temperature metrics in the method early and**
**[2] Revisions to phrasing and additional considerations to the significance of this method.**
**[3] Revisions to language to clarify methods.**

**AUTHOR RESPONSES TO REVIEWER 2 COMMENTARY ON TUTTON AND WAY MANUSCRIPT**

**[T&W response] We would like to thank Reviewer 2 for taking the time to provide helpful comments on our manuscript. We have responded to each comment below and have made the suggested changes to the final revised manuscript. A short summary of implemented changes is at the bottom of this response.**

*Snow depth is a poorly observed variable: measurement networks are sparse (especially across the subarctic and Arctic) the measurements are prone to uncertainty, and often fail to capture the prevailing landscape-scale variability. Limited tools are available to address these weaknesses, so advancements in low-cost, easy to deploy instrumentation is of high interest to the snow community. This manuscript provides an overview of a new low-cost approach to acquiring snow depth estimates using vertical profiles of light and temperature measurements. The paper is clearly written and provides a careful inter-comparison of different processing techniques to derive snow depth from the profile measurements. While I have only a small number of suggestions on the manuscript, my main concern, as outlined in my first comment below, is with the experiment design.*

*1. Because no independent time series of snow depth measurements were acquired coincident to the profile measurements, it is not possible to know what the 'true' snow depth was. This means the various approaches to estimating snow depth can be compared, but not assessed. Only in the case of Goose Bay can the SCLT-derived snow depth time series be shown alongside an independent measurement, and as rightly pointed out in the text, this comparison serves to highlight the differences between snow measurements made in open environments (airports) versus in the forest. Ideally, an SR-50 or another independent sensor would have operated alongside the SCLT profile, although I understand the cost of such a deployment could be prohibitive. To mitigate the lack of assessment and highlight the inter-comparison, it would be useful to see the multiple snow depth time series produced at each location from the various techniques on a single plot (individual panel for each site). This would illustrate the range/agreement in snow depth through time based on the analysis methodology, which can now only be inferred by flipping between figures and looking at the correlation results in Figure S3. I suggest adding this new figure, and a brief discussion of it, to Section 5.*

**[T&W response] We agree and acknowledge that the lack of validation poses some challenges in evaluating the method. We have added an additional figure which compares the snow depth results for each of our SCLT sites to the CMC snow dataset (Brown et al., 2003; Brown and Brasnett, 2010) (Section 5.1). Although these data are based on a combination of first-guess modelling and regional weather stations, this product is widely-used and does represent variability in snow characteristics for an overlapping period with our stations. The new comparison we have added (Fig. 12) shows reasonable agreement between snow estimates for our sites and the regional CMC product. The comparison also demonstrates a clear difference at BaseSnow but we believe our representation is as likely to be accurate as the CMC product which lacks any observed snow depth inputs in this region.**

*2. Line 262-268: consider moving Figure S3 out of the supplemental material to include it in this*

*paragraph. I think these are worthwhile results to include in the manuscript.*

**[T&W response] We agree with this comment and have moved Figure S3 to the manuscript.**

*3. This issue is acknowledged in the Discussion, but why not use consistent vertical spacing of 10 cm? In a relative sense, greater uncertainty with deeper snow is ok, but the current setup dictates that uncertainty will be greater when snow is deeper.*

**[T&W response] There are two reasons for this. The first is that we had deployed ibuttons for many years at these sites using a similar vertical arrangement so for consistency sake we did not want to introduce an even greater inconsistency. The second reason is more practical in that we are often more concerned about the shallower components of the snowpack when considering thermal impacts of snow cover of permafrost in the region therefore this configuration saves costs and serves the original purpose of the stakes. We have amended the text to clarify this point.**

*4. Line 122: can you provide a simple description of the PELT method? Not clear what is meant by 'asymptomatic penalty of 10%'.*

**[T&W response] This penalty coefficient is part of the PELT method cost minimization function (Killick et al, 2011) and optimizes the number of changepoints in a segmentation. An asymptotic penalty is testing for significance, therefore b=0.1 (10%) is indicative of 90% confidence. We agree with the recommendation to provide a simple description of the PELT method in the manuscript with reference to the original derivation of the cost function.**

*5. Line 170: "all SCLT sites except for BaseSnow had a snowpack taller than the 170 uppermost data logger". Murphy's Law at work that there was an unusually deep snowpack during the season that you were evaluating this approach! Do you have a sense of how tall the profile needs to be? Is there any technical limitation to say, a 2 m profile with sensors every 10 cm?*

**[T&W response] This year was a high snow year relative to the 66-year average (Fig. S5). Prior work (e.g. Way and Lewkowicz, 2018) typically did not have snow depths exceeding 160 cm at these sites but as you say, this is an unusual year. There is no reason to limit the stake height to a particular limit other than we had cost, logistical and continuity considerations in the field that led us to choose the heights we did. We have amended the text in one location to reflect this point.**

*6. Cost effectiveness is a major driver of this work, but (unless I missed it) nowhere in the paper is the cost of the SCLT profiles stated. This information would be helpful! What is the cost sensitivity to the vertical resolution of the profile?*

**[T&W response] Cost of SCLT profiles is presented in Table S1 in the supplemental materials and compared to common iButton stakes. The cost is linear relative to the number of loggers installed, where total cost = installation cost + cost per logger * number of loggers. We have amended the text to make more clear reference to this.**

*7. Sections 4.5 and 4.6 provide more detailed analysis of the light intensity methods. To improve the logical structure of the paper, I suggest shifting these up to follow Section 4.3, and shift down the temperature measurement approach reported in Section 4.4.*

**[T&W response] We agree with moving Section 4.5 up to follow Section 4.3; however, we have kept Section 4.6 in place as it follows the presentation of the temperature method.**

*8. The reference list needs to be cleaned up. Some citations are missing (e.g. Archer, 1998) and details are missing from some references (journal titles, etc.). Review the sequence of figure numbers: figures jump from 10 to 13 to 15.*

**[T&W response] We agree and have addressed the errors in the reference list. Figure order was revised after editor comments and with the reviewer's comments above. It should be noted that referencing issues were introduced through our use of the .csl file for The Cryosphere in the public repository.**

*9. While the code and data availability are provided, what about schematics to the design of the probe? Will these be shared in some form so that others can follow your design if interested?*

**[T&W response] We have provided a diagram of the stake setup in Figure 2 and Figure 3. If readers request further instruction on installation we can provide a detailed technical drawing upon request.**

*Editorial*
*Line 30: change to 'snowfall is hard to catch, melts differentially once on the ground...''*

**[T&W response] We agree and have made this change.**

*Line 55: not clear what is meant by 'relatively unambitious method'. . .uncomplicated?*

**[T&W response] We have changed relatively unambitious method to direct method for clarity.**

*Line 60: suggest changing to 'broader snow science community'*

**[T&W response] We agree and have made this change.**

*Line 118: change to "We determined SCLT-derived snow surface heights using. . ."*

**[T&W response] We agree and have made this change.**

*Figure 4: minor point, but the y-axis range for BaseSnow is slightly different from the other sites*

**[T&W response] We agree and have changed the axis to be consistent.**

**SUMMARY OF CHANGES:**
**[1] Included further validation to CMC snow depth analysis dataset and comparison between methods in manuscript.**
**[2] Added brief background for using stake height and logger distribution.**
**[3] Included additional description of the statistical methods used in the changepoint analysis.**
**[4] Included further reference to the cost differences described in Table S1.**
**[5] Made revisions to missing references and errors in citations.**
**[6] Made revisions to language to improve clarity of manuscript.**

**A low-cost method for monitoring snow characteristics at remote field sites**

Rosamond J. Tutton[1], Robert G. Way[1]

[revised manuscript text omitted]

In this study, we present results from a novel low-cost technique for snow depth estimation that can be efficiently applied at remote field sites. Within a similarsimilar per site cost budget (Table S1), tThe method we propose alleviates some of the challenges associated with other low-cost methods while offering a direct t relatively unambitious straight forwamethod

60 of estimating snow characteristics in natural conditions. Building on the practice of using temperature loggers (Danby and Hik, 2007; Lewkowicz, 2008), we propose the snow characterization with light and temperature (SCLT) technique which uses vertically arranged dual light & temperature data loggers together to produce reliable estimates of snow characteristics with minimal analysis across ecotones. The current generation of SCLT-based snow thickness estimation relies most on light measurements but SCLT's dual sensor configuration will enable future use of multivariate statistical techniques to improve

65 methods prioritize characterization by light, however temperature is used to define thresholds and will be further incorporated in future revisions. . 
[revised manuscript text omitted]

---

## Editor Decision (ED1)

Dear Ms Tutton and Dr Way,

Thank you for your response to reviews. I must apologise for the delay in responding to your comments. I have reviewed the responses and would now like to invite you to proceed with the upload of your revised manuscript.

As well as the comments from the reviewers, to which you have detailed your responses, I also request the following amendments:

Please could you revise the first sentence in the abstract: it is now rather overlong and/or lacking commas

L14: no need for a hyphen in 'one-year'

A close up photo of the loggers mounted onto the stakes would be a great addition to Figure 3 – the general set up can be seen, but a close up of the sensors and the mounting would be beneficial to readers.

Thank you for your patience with the review process; I look forward to reading the next version of the manuscript.

Dr Liz Bagshaw

Editor

---

## Author Response (AR2)

**AUTHOR RESPONSES TO EDITOR COMMENTARY ON TUTTON AND WAY MANUSCRIPT**

*Line 11: remove 'in this study' – it's not necessary since the rest of the sentence is in the first person*

**[T&W response] We thank the editor for these minor revisions and have removed 'in this study'.**

*Line 63-65: 'The current generation of the SCLT-based snow thickness estimation relies most on light measurements but SCLT's dual sensor configuration will enable future use of multivariate statistical techniques to improve snow estimation' – could you define what aspect of 'snow estimation' – depth, water, temperature?*

**[T&W response] We agree with this comment and have specified 'snow-depth estimation' for clarity.**

**SUMMARY OF CHANGES:**
**[1] Removed unnecessary wording from line 11**
**[2] Specified type of snow estimation (depth) on line 64**
**[3] Replaced Figure 1 with a new map that include latitudinal and longitudinal suffixes and included (a), (b), etc, notation for simple reference and adjusted caption considering this change**
**[4] In Figure 2 changed text within the image from "Snow Pack" to "Snowpack"**
**[5] Replaced ";" with "," in the caption for Figure 3 for consistency with the rest of the captions**

[revised manuscript text omitted]